# Review: Inelastic Constitutive Modeling: Polycrystalline Materials [note 1]

**DOI:** 10.3390/ma16093564

**Published:** 2023-05-06

**Authors:** Mirza Baig, Josiah Owusu-Danquah, Anne A. Campbell, Stephen F. Duffy

**Affiliations:** 1Department of Civil and Environmental Engineering, Washkewicz College of Engineering, Cleveland State University, Cleveland, OH 44115, USA; 2Materials Science and Technology Division, Oak Ridge National Laboratory, Oak Ridge, TN 37831, USA

**Keywords:** continuum mechanics, constitutive modeling, time dependent behavior, inelastic deformations

## Abstract

This article provides a literature review that details the development of inelastic constitutive modeling as it relates to polycrystalline materials. This review distinguishes between inelastic constitutive models that account for nonlinear behavior at the microstructural level, time-independent classic plasticity models, and time-dependent unified models. Particular emphasis is placed on understanding the underlying theoretical framework for unified viscoplasticity models where creep and classical plasticity behavior are considered the result of applied boundary conditions instead of separable rates representing distinct physical mechanisms. This article establishes a clear understanding of the advantages of the unified approach to improve material modeling. This review also discusses recent topics in constitutive modeling that offer new techniques that bridge the gap between the microstructure and the continuum.

## 1. Introduction 

Nonlinear hereditary inelastic deformation behavior can occur in many materials utilized at elevated service temperatures. This behavior can include creep, rate sensitivity, and plasticity. Accurate assessments of nonlinear stress and deformation behavior are important in predicting the operational life and overall performance of critical engineering systems. Inelastic constitutive models have been developed and deployed to meet these assessment needs. The models must predict nonlinear behavior under complex thermomechanical load paths. This includes capturing phenomena such as Bauschinger’s effect, cyclic softening and hardening, stress relaxation, and ratcheting when present in high-temperature applications.

In this article, focus is given to theoretical concepts that historically supported the development of unified inelastic constitutive models. The field is rich, and because of this, there will be some oversights here, and apologies are offered a priori. The discussion begins by mentioning the work of Andrade [1,2] with soft metals (e.g., lead). Andrade modeled transient creep strain by assuming it is proportional to the cubic root of time. Expressions for different stages of creep (primary, steady state, but not tertiary) have been offered by Norton [3] and Bailey [4]. The generalization of Norton’s power law to multiaxial states of stress using a potential function is highlighted later in the viscoplasticity section of this paper. Odqvist [5] published a multiaxial formulation for steady-state (secondary) creep prior to the modern-day work on unified viscoplasticity models, and he later extended the model to include primary creep [6]. However, as indicated later, experimental evidence supports the use of unified models in modeling the first two stages of creep as opposed to a piecemeal approach. In addition, the interaction between creep and plasticity must be addressed in a unified fashion. Nonetheless, some attention has been given here to certain material science models that explain nonlinear behavior at the microstructural level. An understanding of material microstructure is always necessary to develop multiaxial continuum-level constitutive models. Reaction-rate and power-law formulations for microstructural constitutive relationships are discussed. The work in developing rheological viscoelastic constitutive models by Schapery [7] and Findley et al. [8] is briefly presented since they point the way to the endochronic viscoplastic models that appear later. The objective of this article is to highlight the genesis of unified time-dependent viscoplastic models.

Early work by Besseling [9] suggested separate strain rates for plasticity and creep that were based on partitioning a total strain rate. Specifically, he proposed a one-dimensional model where the total strain rate could be partitioned as follows:(1)$$\frac{d\varepsilon^{T}}{dt}=\frac{d\varepsilon^{E}}{dt}+\frac{d\varepsilon^{C}}{dt}+\frac{d\varepsilon^{P}}{dt}$$
where

*ε^T^—*Total strain

*ε^E^—*Elastic strain

*ε^C^—*Creep strain

*ε^P^—*Plastic strain

The inelastic components of this expression are as follows:(2)$$\begin{array}{cc} \frac{d\varepsilon^{I}}{dt} & =\;\mathrm{Inelastic}\;\mathrm{Strain}\;\mathrm{Rate}\;\\ & =\frac{d\varepsilon^{C}}{dt}+\frac{d\varepsilon^{P}}{dt}\\ & ={\overset{˙}{\varepsilon }}^{I}\left(\varepsilon^{C},\varepsilon^{P},t\right) \end{array}$$
where the two rates representing creep and plasticity are identified as separate and distinct deformation mechanisms.

The equations above are associated with microstructural creep mechanisms that are thermally activated. For materials utilized in fabricating nuclear reactor components that function as moderators, e.g., graphite, additional creep mechanisms arise that are activated by irradiation. As Onimus et al. [10] point out, Equation (2) should be modified as follows:(3)$$\begin{array}{cc} \frac{d\varepsilon^{I}}{dt} & =Inelastic\;Strain\;Rate\\ & =\frac{d\varepsilon^{P}}{dt}+\frac{d\varepsilon^{\mathrm{thermal}-\mathrm{creep}\;}}{dt}+\frac{d\varepsilon^{\mathrm{irradiation}-\mathrm{creep}\;}}{dt}+\frac{d\varepsilon^{\mathrm{swelling}/\mathrm{growth}\;}}{dt}\\ & ={\overset{˙}{\varepsilon }}^{I}\left(\varepsilon^{P},\varepsilon^{\mathrm{thermal}-\mathrm{creep}\;},\varepsilon^{\mathrm{irradiation}-\mathrm{creep}\;},\varepsilon^{\mathrm{swelling}/\mathrm{growh}\;},t\right) \end{array}$$

Expressions for creep strain rates are identified as *ε*^*thermal*-*creep*^, *ε*^*irradiation*-*creep*^, and *ε*^*swelling*/*growth*^ in publications cited above or based on models established through material science concepts. The forms for creep strain rates based on material science are dependent on various facets of a material microstructure. For example, dislocation mechanics include models for dislocation glide, dislocation creep, and diffusional flow. These mechanisms and the attending creep strain models are discussed in the following section.

For the plasticity component of Equation (2), incremental theories are deployed. Classic plasticity theory poses a constitutive relationship that is independent of the rate of loading. However, plasticity is path-dependent, and load sequences are often represented as a process evolving in time. Hence, classical plasticity theories are posed on an incremental basis, and Equation (2) would be better stated in a differential format:(4)$$d\varepsilon^{I}=d\varepsilon^{P}+d\varepsilon^{\mathrm{thermal}-\mathrm{creep}\;}+d\varepsilon^{\mathrm{irradiation}-\mathrm{creep}}+d\varepsilon^{\mathrm{swelling}/\mathrm{growth}}$$

Equation (4) can be easily integrated with respect to time provided that the load path considered can be expressed as increments of time. Details associated with incremental plasticity models are presented in a later section.

The modeling of plasticity and creep can be accomplished separately, or represented as a single rate, i.e., as a unified model. Nonunified models allow creep strain to be rate-dependent, but plastic strains on the other hand are rate-independent. Alternatively, in unified constitutive models, creep and plastic strains are represented by a single inelastic strain measure that is considered inherently rate-dependent. Nonunified models lack the ability to predict interactions between creep and plasticity, e.g., ratcheting (cyclic creep) and stress relaxation. Experimental evidence such as accumulated creep strain affecting the size of the threshold (yield) surface and limited creep strain recovery during the unloaded periods accompanied by significant reductions in the resistance to creep upon subsequent reloading have been observed that cannot be captured by nonunified models. Jaske et al. [11] as well as Pugh and Robinson [12] have pointed out that the behavior of metallic alloys under cyclic load at elevated service temperatures is indicative of an interaction going on between the creep and plasticity components of Equation (2). Many authors, including Robinson [13,14,15,16,17,18,19,20,21] have pointed out that separating creep and plasticity into distinct strain rates as is performed in nonunified models leads to an inability to capture the interactive behavior exhibited experimentally. Phenomenologically based unified creep–plasticity interaction models were developed from theoretical frameworks based on a single inelastic rate ${\overset{˙}{\varepsilon }}^{I}$. The unified manner of representing a constitutive relationship enables a comparison of various aspects of the model to nonlinear behaviors. The details of a number of these unified models are presented later.

The dichotomy between classical mechanisms (plasticity and material science creep) associated with Equation (2) and unified modeling approaches is presented in Figure 1. In that overview figure, inelastic constitutive models are first segregated into three sub-categories: models that explain nonlinear behavior at the microstructural level, time independent classic plasticity models, and time-dependent unified models. The models where the time-dependent inelastic strain is derived from a single potential or a single integral function are classified as unified models.

Plasticity, as well as thermally activated material science creep models, and creep models based on mechanism activated by irradiation are discussed separately in the following two sections for historical completeness. The remainder of the article primarily focuses on unified viscoplasticity models where creep and classical plasticity behavior are considered the result of applied boundary conditions instead of separable rates representing distinct physical mechanisms. The existing models for a unified strain rate (e.g., Robinson et al. [13,14,15,16,17,18,19,20,21], Chaboche et al. [22,23,24], Bodner et al. [25,26,27,28,29,30,31,32,33,34,35], Miller et al. [36,37,38,39,40,41,42,43,44,45,46,47,48] and Walker et al. [49,50,51]) are based on constructs involving internal state variables that capture several inelastic hardening behaviors. Chan et al. [52], as well as Allen and Harris [53], provide overviews of the viscoplastic models just mentioned.

There are micromechanics models for polymer composite material systems such as the models proposed by Chamis [54]. Here, each component of the microstructure is treated as a continuum. These models are discussed elsewhere, and the rest of the article focuses on models that capture the macro-level response of materials exhibiting inelastic behavior.

## 2. Early Efforts in Modeling Creep Behavior

The Norton–Bailey [3,4] creep model mentioned earlier is a power law formulation that is not associated with a specific microstructural mechanism. In this effort, the Norton–Bailey [3,4] model is categorized as a material science model and is included with other microstructural creep strain models since the model is deployed both at the microstructural level and the continuum level (although the model is one-dimensional). The model was derived predicated on the assumption that creep strain is functionally dependent on stress, time, and temperature. The functional form for the Norton–Bailey [3,4] creep model is based on the following separable formulation
(5)$$\varepsilon^{C}=F_{1}\left(\sigma \right)F_{2}\left(t\right)F_{3}\left(T\right)$$

Several forms for *F*_1_ can be utilized, including Norton’s [3] model
(6)$$F_{1}\left(\sigma \right)=A\sigma^{m}$$
an exponential format
(7)$$F_{1}\left(\sigma \right)=Cexp[\frac{\sigma }{\sigma_{0}}]$$
and a hyperbolic sine format
(8)$$F_{1}\left(\sigma \right)=sinh^{m}[\frac{\sigma }{\sigma_{0}}]$$

The hyperbolic function is convenient since at the limits for low-stress values, the function approaches values given by the power law, and at high-stress values, the hyperbolic takes on values nearly equal to the exponential function.

The time function *F*_2_(*t*) is also formulated as a power law in time
(9)$$F_{2}\left(t\right)=at^{n}$$

Equation (9) is attributable to Bailey [4]. Finally, the temperature function is stipulated as
(10)$$F_{3}\left(T\right)=F_{3}[\exp(\frac{-Q}{RT})]$$

In this expression,

*Q* = activation energy

*R* = Boltzmann’s constant

Functions *F*_2_ and *F*_3_ are combined such that
(11)$$F_{4}\left(t,T\right)=\alpha [t\exp\left(\frac{-Q}{RT})\right]$$
and
(12)$$\varepsilon^{C}=A\sigma^{m}[t\exp(\frac{-Q}{RT}){]}^{n}$$

Under isothermal conditions (*T = constant*),
(13)$$\varepsilon^{C}=A\sigma^{m}t^{n}$$

Under the assumption that the length of exposure time to temperature and stress level are the major factors affecting the creep strain rate,
(14)$$\begin{array}{cc} \varepsilon^{C} & =F_{1}\left(\sigma \right)F_{2}\left(t\right)\\ & =\left\{A\sigma^{m}\right\}\left\{\alpha t^{n}\right\}\\ & =\alpha A\sigma^{m}t^{n} \end{array}$$

Taking a time derivative of Equation (14) leads to a rate formulation of the Norton creep law
(15)$$\frac{d\varepsilon^{C}}{dt}=\overset{}{\overset{\sim }{A}}\sigma^{m}t^{\overset{}{\overset{\sim }{n}}}$$

Note that the constants
(16)$$\;\overset{}{\overset{\sim }{A}}=\alpha An$$
and
(17)$$\overset{}{\overset{}{\overset{\sim }{n}}}=n-1$$
are assumed to be temperature-dependent.

## 3. Creep Strain Models Based on Microstructural Mechanisms

When considering continuum-level deformation for polycrystalline materials, it is important to be cognizant of the microstructural mechanisms driving the deformation processes. There are two primary mechanisms that drive creep strain rates. The first mechanism is diffusional creep (Coble creep [55], as well as Nabarro [56] and Herring [57] creep) where vacancies are driven through the crystal lattice of the material. The second process involves the translation of dislocations through the atomic matrix of the material (see Weertman [58]). For either mechanism, the overall stiffness of the material decreases with time. Stress–strain curves that are initially linear in the elastic range bend over as total strains accumulate. The decrease in stress represents the hardening of the material when subjected to monotonic loads. The decrease in strain rate to a steady-state value in a material subject to a constant load represents two material mechanisms competing and eventually balance a hardening response with a recovery response.

Figure 2 represents a generic deformation mechanism map. In this figure, dominant microstructural mechanisms are mapped to various load and temperature regimes. At low homologous temperatures, i.e., <(0.5) T/T_m_, where T is the service temperature and T_m_ is the material melting temperature, deformations are primarily elastic until high-stress regimes are attained. At moderate temperatures and high stress, dislocations tend to glide on slip planes. Contrasted with other diffusional creep mechanisms, Coble creep [55] is similar to Nabarro–Herring creep [56,57] in that both are dominant at lower stress levels and higher temperatures. Dislocation glide mechanisms occur at higher strength levels at all temperatures. Coble creep [55] occurs through the diffusion of atoms in a polycrystalline material along grain boundaries. At elevated temperatures, dislocation mechanisms are aided by vacancy diffusion. Vacancy diffusion also causes deformations without any dislocation movement involved.

Models exist that attempt to bridge the gap between microstructural-level modeling and continuum-level modeling. These include the rate-dependent deformation models of Ponter and Leckie [59]. Here, a simple constitutive framework incorporates hardening and recovery mechanisms associated with dislocation motion. Similarly, Rice [60] models dislocation motion using the normality structure which has served many times as a fundamental principle in continuum-level macroscopic constitutive laws. In Rice’s [60] model the local microstructural rearrangements proceed at a rate governed by a thermodynamic force.

## 4. Irradiation-Induced Creep

Another form of plastic deformation is irradiation-induced creep, which occurs in materials exposed to radiation under applied stress. Irradiation-induced creep occurs at temperatures well below 50% of the melting temperature of a material because the collisions of the radiation with the lattice atoms produce the defects (self-interstitials and lattice vacancies) that allow creep to occur rather than relying on thermal lattice vibrations to produce the creep. The mechanisms of diffusional creep (i.e., Coble creep [55] and Nabarro–Herring creep [56,57]) are not enhanced by irradiation [61,62]. Instead, the primary mechanisms of irradiation creep are the motion of dislocations, i.e., either enhanced climb or climb and glide of dislocations, and the preferential formation of loops. Stress-induced preferential absorption (SIPA) occurs when the climb of dislocations is accelerated by preferential absorption generally of interstitials. Stress-induced climb and glide (SICG) occurs when irradiation defects allow dislocations to climb over pinning points and the subsequent glide is the source of the creep strain. When the applied stress affects the climb velocity (i.e., SIPA with SICG), the resulting behavior is referred to as the preferred absorption glide (PAG). Stress-induced preferential nucleation (SIPN) occurs when dislocation loops nucleate in the crystal structure with a preferred orientation.

Creep of alpha-uranium was observed in the 1950s [63]. The mechanism proposed was that internal stresses due to anisotropic growth of the crystals, when combined with externally applied stress, can overcome the yield strength, thereby causing plastic deformation. However, Hesketh [64] instead proposed a mechanism that depends on the anisotropic growth of crystals, the SIPN of dislocation loops, and the climb of the SIPN loops. In austenitic stainless steels and nickel-based alloys, primary creep is mainly due to dislocation slip (SICG) that is reduced as the material hardens (increased pinning point density as fluence increases), while steady-state creep is thought to be primarily due to SICG and SIPA, but the mechanism can change as experimental parameters change [62]. Ferritic–Martensitic steels undergo creep, but SIPA and PAG are not sufficient to describe the total creep, so it is accepted that SIPN is also an active mechanism [62]. In zirconium alloys, irradiation creep is thought to be due to both thermal dislocation slip with irradiation-enhanced climb and SIPA. Cubic silicon carbide (3C-SiC) irradiation creep has a stress exponent near one, indicating a SIPA mechanism, but microscopic studies have also observed an increase in the number of loops in the {111} planes, indicating some influence of SIPN [65].

Irradiation creep in graphite has been studied quite heavily, yet there is still only limited agreement on the microstructural mechanisms. Primary creep in graphite is saturated around one elastic strain unit, defined as applied stress divided by Young’s modulus and is primarily thought to be driven by dislocation bowing [66,67,68] and interactions with pinning points. The secondary creep (i.e., steady state) mechanism was originally proposed to be driven by anisotropic grain growth, but that was shown to not be the case when boron additions changed the rate of grain growth but did not change the creep rate [69]. The long-accepted mechanism is the pinning–unpinning mechanism proposed by Kelly and Foreman [70]. Strain in this mechanism is driven by dislocation glide, and the rate is controlled by the creation and destruction of 2-4 atom clusters that act as the dislocation pinning points. With this being a glide-based mechanism (SICG), it should have a stress dependence of two. Newer published irradiation creep results [66,71,72,73] suggest a SIPA mechanism. Results from Kennedy in the 1960s [66] showed that changing stress during irradiation did not affect the primary creep strain, which provides additional results that disagree with the pinning–unpinning mechanism and instead support a SIPA model.

## 5. Time-Independent Models Based on Continuum Principles: Plasticity

Historically, the classic plasticity models with flow theories and hardening laws have been proposed to model material time-independent inelastic behavior, i.e., plasticity. The theoretical basis for yielding under complex states of stress at a continuum level stretches back to Coulomb’s [74] effort in defining failure using his nascent concepts in soil mechanics. Yielding in the classic plasticity sense had its origins in the nineteenth century when Tresca [75] conducted a series of punching and extrusion experiments that led him to postulate that yielding occurred when the maximum shear stress at a point reached a critical value.

Von Mises [76] developed a mathematically based yield criterion, which was later explained by Henry [77] as plastic yield when the elastic shear strain energy at a point reached a critical value. Von Mises [76] separately published equations for rigid-perfectly plastic materials, which resembled Levy’s [78] work. In 1913, Von Mises [76] proposed his widely accepted *J*_2_ yield criterion that stipulated that yielding took place when critical shear values were attained on octahedral planes (see Nadai’s work [79]). Huber [80] had earlier published essentially the same criterion. In 1948, Hill [81] proposed the first anisotropic yield criterion. Duwez [82] established a stress–strain relationship for plasticity in single-crystal materials at the microstructural level. In that work, the existence of a “secondary structure” in crystals was adopted, where the material is assumed to be composed of *N* parallel sub-elements. This model was later modified by Besseling [9].

In Figure 1, time-independent plasticity theories are loosely grouped into models based on flow theories and endochronic theories. Plastic flow theories [83,84,85] laid the foundation for the formulation of constitutive equations for plasticity modeling. The 1940s saw the advent of the classical theory; Hill [81], Koiter [83], Prager [84], and Drucker [85], among others, brought together many fundamental aspects of the theory into a single focus. The endochronic theory of plasticity, an intrinsic time theory proposed by Valanis [86,87,88,89] without a yield surface, was able to capture the phenomena of cyclic hardening, initial strain problems, and cross-hardening, unlike classical plasticity. Here, the same constitutive equation is applied for loading and unloading processes; Bazant and Bhat [90] showed that endochronic theories can capture inelasticity and failure in concrete.

Models formulated at a continuum level in a multiaxial setting are useful to engineers designing components in high-temperature applications (nuclear reactors, combustor sections of jet engines, etc.). In continuum theories, stress, strain, and other field quantities are posed at a mathematical point. In applying constitutive models to structural components with internal microstructures, a continuum point must be reinterpreted as a continuum element of finite dimension. The continuum element must be small enough to be nearly homogenous in the presence of stress and strain, as well as temperature gradients, but large enough relative to the microstructure being homogenized. Based on these concepts and well-chosen phenomenological experiments, a system of constitutive relationships can be formulated in support of conducting structural analyses for engineering design. Thus, in this section and succeeding sections, constitutive models have been reviewed that help design components, not the material. Microstructural constitutive models focus more on designing the material. At the continuum level, one designs with the material. At the microstructural level, one designs the material.

Deformation analyses proceed on an incremental basis in classic theories of time-independent plasticity. One calculates increments in plastic strain (*dε*^*P*^_*ij*_). There are three essential elements to a work hardening-based incremental inelastic constitutive law. First is the existence of a threshold function delineating elastic states of stress—a yield criterion. The threshold function typically serves as a potential function. Consider the yield criterion attributed to Tresca [75], von Mises [76], and Hill [81], all of which are stipulated at the continuum level. The second element is a hardening rule, also referred to as an evolutionary law. A hardening rule mathematically describes the evolution of a potential function (i.e., how a material “hardens”) to accommodate the effects of inelastic deformation. The third element is the flow rule. Mendelsohn [91] as well as Chen and Han [92] describe the link between incremental plastic strain and incremental changes in the state of stress captured by the flow rule. Inelastic modeling is primarily based on two types of flow rules. The first is referred to as the associated flow rule, where the threshold function behaves like a potential function. Historically constitutive models for ductile metals are modeled based on this associated flow rule (see Equation (18) below).

The second type is known as a non-associated flow rule, where the inelastic strain rate vector is not normal to the threshold surface. Associated flow rules are more prevalent, so this type of constitutive model will be focused on.

Since classical time-independent incremental plasticity models assume that increments in plastic strains (*dε*^*P*^_*ij*_) are normal to a yield surface. Flow rules, often referred to as normality rules, take the following form:(18)$${d\varepsilon^{P}}_{ij}=\lambda \left(\frac{\partial f}{{\partial \sigma }_{ij}}\right)\;\lambda >0$$

Thus, an increment in plastic strain is coincident with the gradient of the scalar-valued flow potential function, *f*. Here, *f* represents the yield function that is dependent on the stress tensor *σ*_*ij*_ such that
*f* = *f* (*σ*_*ij*_, *H_α_*)
(19)
and it incorporates a state variable vector *H_α_*. This vector can contain a single state variable, i.e., *K*, for isotropic hardening; six state variables associated with kinematic hardening and represented by the symmetric tensor *α*_*ij*_; or seven if both isotropic hardening and kinematic hardening are present, i.e.,

*H_α_*_= 1, ⋯, 6_ = *H* (*K*, *α*_*ij*_)
(20)

A loading rule must be established before inelastic (plastic) strains can be quantified. The loading rule determines whether inelastic (plastic) strains occur along an incremental load path. Since the yield (threshold) function identifies elastic states of stress, stress states outside of the surface of the threshold function are mathematically inaccessible. States of stress within the threshold surface represent elastic states of stress. The inaccessible states of stress outside the boundary of the yield function can be subsequently absorbed within the surface by evolving the boundary of the threshold function as plastic strains accumulate. This evolution process eventually migrates to the functional boundary sufficiently so that stress states beyond the boundary are subsumed. The threshold-potential function therefore must be dependent on stress, as well as the appropriate internal state variables.

A change in the stress state that does not change the inelastic state variable vector *H_α_* corresponds to unloading into or around (tangent to) the elastic stress region. As unloading takes place,
*df* < 0
(21)
and the value of the components of the inelastic state variable vector scalar does not change. When a material is loaded such that a state of stress is on the inelastic threshold surface (*f* = 0) and an increment in stress, *dσ*_*ij*_, is applied such that
(22)$$\left(\frac{\partial f}{{\partial \sigma }_{ij}}\right)d\sigma_{ij}>0$$
then the increment in stress gives rise to an increment in inelastic strain and a corresponding change in the inelastic state of the material, i.e.,

*dH_α_* ≠ 0
(23)

This change in inelastic state corresponds to the material hardening and in turn impacts the stress–strain curve. The presence of inelastic strains can be detected through the nonlinear behavior of the stress–strain curve or by unloading the material and noting the permanent strains. In general, the change in an inelastic state variable follows the following tensor expression

*dH**_α_* = *G_ij_**_α_*  
*dε^P^_ij_*
(24)

The incremental stress *dσ*_*ij*_ evolves the threshold function. The inelastic behavior of a material is best described mathematically by
(25)$$\begin{array}{c} d\varepsilon_{ij}^{p}\neq 0\\ dH_{\alpha }\neq 0 \end{array}\left\{\begin{array}{c} f\left(\sigma_{ij},H_{\alpha }\right)=0\\ and\\ \left(\frac{\partial f}{\partial \sigma_{ij}}\right)d\sigma_{ij}>0 \end{array}\right.$$
or
(26)$$\begin{array}{c} d\varepsilon_{ij}^{p}=0\\ dH_{\alpha }=0 \end{array}\left\{\begin{array}{c} f\left(\sigma_{ij}+d\sigma_{ij},H_{\alpha }\right)<0\\ or\\ f\left(\sigma_{ij},H_{\alpha }\right)=0\\ and\\ \left(\frac{\partial f}{\partial \sigma_{ij}}\right)d\sigma_{ij}<0 \end{array}\right.$$

Specific forms for *dε*^*P*^_*ij*_ and *dH_α_* depend on the potential function adopted and the hardening laws assumed for the material as inelastic deformations take place. Note that the form of the inelastic potential function can be utilized to extract different types of anisotropic behavior from a plasticity model, as well as different behavior in tension and compression (see Green and Mkrtichian [93]). The theoretical framework for classic plasticity provided above serves as a frame of reference for the time-dependent viscoplastic models discussed next.

## 6. Time-Dependent Isotropic Unified Models: Viscoplasticity

Predicting time-dependent stress depends not only on the current strain state but also on the previously accrued strains, i.e., the material strain history. Two approaches can be taken in characterizing nonlinear hereditary stress–strain behavior. They involve solving either a system of differential equations or a system of integrals. The first, discussed in detail here, is referred to as internal-state variable representation. Horstemeyer and Bamman [94] provide an excellent historical overview of the internal state variable approach to modeling constitutive behavior. The state variable method develops a system of differential equations that quantify an inelastic strain rate as well as differential equations that govern the evolution of the state of the material during deformation. The viscoplastic model based on a system of differential equations requires a flow rule, an evolutionary law to allow for changes in the state variables adopted, and thermodynamic stability constructs. Adopting a state variable representation is a convenient and flexible thermodynamic format for the description of nonlinear hereditary behavior. The second approach develops a system of integrals to account for stress–strain history. The integral approach, or endochronic theory, represents the inelastic behavior of materials using the notion of intrinsic time. The formulation of a flow rule in endochronic models is not predicated on the use of flow potentials. A third approach is a hybrid approach based on characterizing specific changes in a material’s microstructure during the deformation process. Finite element analysis is employed to develop a representative volume element (RVE) at the microstructural level. Models based on these RVEs can then be used to analyze components at the continuum level once the RVE model is constructed. However, the microstructural models tend to ignore recovery mechanisms that appear explicitly in continuum-level modeling. With the RVE approach, an Arrhenius function and reaction rate theory is applied to a slip process of choice to quantify a strain rate. This third approach will be addressed later in the discussion on how to bridge microstructural concepts and continuum concepts.

As Ponter and Leckie [59] point out, the internal state variable method mirrors many elements of classical plasticity presented in the previous section. They also indicate that internal state variable models are a marriage of classical plasticity concepts with the Norton [3] creep law outlined earlier. A flow law is required to quantify the inelastic strain rates. Rice [60] outlined thermodynamic justification for the existence of a flow potential from which inelastic strain rates are obtained. Ponter and Leckie [59] generalized Norton’s [3] creep law to viscoplasticity for multiaxial states as
(27)$$\frac{d\varepsilon_{ij}^{I}}{dt}=\frac{\partial }{\partial \sigma_{ij}}\left\{\frac{\phi^{n+1}}{n+1}\right\}$$
where the bracketed term represents Rice’s [60] scalar flow potential function, denoted as *Ω* in the literature. An evolutionary law (see below) is needed to characterize the change in the internal state variables used to characterize the material microstructure. A third element of an internal state variable model is a criterion stipulating when inelastic deformations will take place. Here, it will be referred to as a threshold function. The threshold function delineates regions of elastic behavior and inelastic behavior in the six-dimensional stress space. For viscoplastic constitutive models based on the internal-state variable method, the threshold function can serve as a flow potential function in Equation (27) above. Threshold functions are key in deriving growth laws that capture various types of hardening mechanisms (e.g., isotropic, kinematic, irradiation) using the internal state variables adopted for a particular model.

The governing differential equations of an internal state variable formulation for viscoplasticity are associated with the normality structure of the threshold potential function *Ω*. Consider the following the dependence for the threshold potential function suggested by Ponter and Leckie [59]:
*Ω* = *Ω* (*F*, *G*)
(28)
where *F* is dependent on both the stress state (*σ_ij_*) and the inelastic state variable (*α_ij_*), i.e.,

*F* = *F* (*σ_ij_, α_ij_*)
(29)

The second function *G*
*G* = *G* (*α_ij_*)
(30)
is only dependent on inelastic state variables, which are represented by a second-order tensor. Now
*Ω* = *Ω* (*σ_ij_, α_ij_*)
(31)
and the flow law according to Rice [60] is
(32)$${\overset{.}{\varepsilon }}_{ij}^{I}=\frac{\partial \Omega }{\partial \sigma_{ij}}$$

Ponter and Leckie [59] stipulated
(33)$$\overset{.}{\alpha_{ij}}=-\frac{1}{h\left(\alpha_{ij}\right)}\frac{\partial \Omega }{\partial \alpha_{ij}}$$
where *h* is a scalar-valued function of the internal state variables only. Equations (32) and (33) stipulate the normality structure of the inelastic strain rate and the rate of change in the inelastic state variable with the threshold potential function *Ω*.

Based on the dislocation mechanics work of Orowan [95], creep is a competitive process between work hardening, where dislocations in the microstructure interact and local dislocation densities increase and dynamic recovery mechanisms reduce the density of dislocations. Dynamic recovery is a thermally activated mechanism in which immobile dislocations that have piled up are released and the stored energy in the microstructure is diminished. This release corresponds to an obvious decrease in dislocation density. When these two mechanisms balance, the result is a steady state condition. Mitra and McLean [96] suggested how to verify Orowan’s [95] competing process concept experimentally. From a uniaxial stress perspective consider

*σ* = *σ* (*t*, *ε*)
(34)

Taking the differential of this relationship leads to
(35)$$\;d\sigma =\left(\frac{\partial \sigma }{\partial t}\right)dt+h\left(\frac{\partial \sigma }{\partial \varepsilon }\right)d\varepsilon \;=-Rdt+Hd\varepsilon$$

Here, *R* is the magnitude of the recovery rate and *H* is a work-hardening coefficient. Note that stress is constant when the material undergoes steady state creep. Thus, for steady state creep conditions,

*dσ* = 0
(36)

As a result, the following relationship can be obtained from Equation (35)
(37)$$0=-Rdt+Hd\varepsilon \;{\left(\frac{\partial \varepsilon }{\partial t}\right)}_{\begin{array}{c} steady\\ state\\ creep \end{array}}=\frac{R}{H}$$

This relationship was originally proposed in Orowan [95]. It took several decades of experimental work to produce the data that support the mathematical framework in Equation (37). The reader is referred to the seminal efforts of Mitra and McLean [96] as well as others (identified in Gan’s [97] overview) for the experimental data supporting Equation (37). In general, the experimental work carefully measured creep strain rates, the rate of recovery (*R*), and the hardening coefficient (*H*) over a range of temperatures and applied stress.

Observations are made in Walker [50] concerning the growth and recovery of the internal state variables due to hardening in the presence and absence of inelastic deformation, respectively. As the growth of the internal state variable due to hardening is dependent on the presence of inelastic deformation, only recovery and annealing changes in the internal state variable are possible during the elastic unloading phase of a thermomechanical cycle. Freed and Robinson [98], Freed and Chaboche [99], and Arnold and Saleeb [100] provide thermodynamic rationales as to the forms of the evolutionary laws for internal state variables. The evolutionary models, along with the flow equations for inelastic strain rate can be utilized to obtain accurate theoretical solutions for a wide variety of time-dependent boundary value problems posed in elevated temperature environments. A general multiaxial formulation for an internal state variable evolutionary law utilized in unified viscoplastic theories takes the following form:(38)$$\frac{d\alpha_{ij}}{dt}=h\left(\alpha_{ij}\right)\frac{d\varepsilon_{ij}^{I}}{dt}-r\left(\alpha_{ij}\right)a_{ij}$$

Given the relationships suggested by Rice [60] in Equation (32), and in Equation (33) suggested by Ponter and Leckie [59], obtaining specific formulations for Equation (38) is strictly dependent on the expression used to characterize the flow potential *Ω*. For each of the viscoplastic models presented in the subsequent sections, the expression for *Ω* is identified. Viscoplastic models constructed based on flow potentials allow for the convenient use of tensorial invariant theory to extend the models and capture a variety of mechanical responses, e.g., anisotropic behavior and different behavior in tension and compression.

Just as in classical plasticity, the unified viscoplastic models reported on here satisfy Drucker’s [85] postulates. In addition, viscoplastic models can satisfy Ponter’s inequalities [101,102], which are stipulated for time-dependent constitutive models. The postulates and inequalities relate to thermodynamic admissibility restrictions outlined in Arnold and Saleeb [100].

## 7. Unified Viscoplasticity Models Based on Potential Functions

Fundamentally, the unified approach treats all aspects of inelastic deformation (plasticity, creep, and stress relaxation) with a consistent set of flow equations and evolutionary equations that track the time-dependent behavior of internal state variables. Viscoplasticity models have been used in the nuclear industry to design reactor components. In the aerospace industry, viscoplastic models are used to design combustor section components and leading-edge technologies in hypersonic applications. This approach is robust and has found widespread use.

Allen and Harris [53] point out that several of the viscoplastic constitutive theories they surveyed had similar elements. The models utilize a set of internal state variables that provide locally averaged representations of microstructure such as dislocation rearrangement and grain boundary sliding. The following multiaxial models make the continuum assumption. Unified constitutive equations can be characterized as mathematically “stiff”. The system of coupled partial differential equations used to solve boundary value problems with viscoplasticity models depends on variables that are susceptible to large changes over small time increments of the independent variables. This “stiff” behavior occurs usually with the onset of a significant amount of inelastic strain in the load step and is due primarily to the nonlinear nature of the functional forms.

### 7.1. Robinson’s Model

Robinson’s model [12,13,14,15,16,17,18,19,20,21] is a unified viscoplastic model based on potential function. This model has well-intended similarities to the structure of the classical plasticity model discussed earlier. The Robinson [15] model possesses a yield criterion (flow potential), and it was originally proposed in a full multiaxial formulation. Under isothermal conditions, the flow potential is dependent on stress, and if kinematic hardening is assumed, the state variable is taken as a deviatoric second-order tensor, i.e., *a_ij_*. For this model,
(39)$$\Omega =K^{2}\left[\left(\frac{1}{2\mu }\right)\int f\left(F\right)dF+\left(\frac{R}{H}\right)\int g\left(G\right)dG\right]$$
where *μ*, *R*, *H,* and *K* are material constants. As the work of Bridgman [103] and others indicates, inelastic deformation is essentially unaffected by the hydrostatic component of the stress state. Robinson [15] originally took stress dependence in terms of the deviatoric components of the applied stress
(40)$$S_{ij}=\sigma_{ij}-\frac{\sigma_{kk}\delta_{ij}}{3}$$
and similarly, the second order deviatoric state variable tensor mentioned is
(41)$$a_{ij}=\alpha_{ij}-\frac{\alpha_{kk}\delta_{ij}}{3}$$

Robinson further identified an effective stress as


Σ*_ij_* = *S_ij_ − a_ij_*
(42)

The dependence upon the deviatoric effective stress S*_ij_* and the deviatoric internal state variable *α_ij_* are introduced through the scalar function
*F* = *F* (Σ*_ij_* )     = *F* (*J*_2_)
(43)
where
(44)$$J_{2}=\frac{1}{2}\sum_{ij}\sum_{ji}$$

Similarly,
(45)$$G=G\left(a_{ij}\right)\;\;=G\left(J_{2}^{\prime }\right)$$
where
(46)$$J_{2}^{\prime }=\frac{1}{2}a_{ij}a_{ji}$$

Finally, for the isotropic version of Robinson’s model, the function *F* takes the following form
(47)$$F=\frac{J_{2}}{K2}-1$$
and
(48)$$G=\frac{J_{2}^{\prime }}{K^{2}}$$

*F* serves as a threshold function. Inelastic strains occur only for states of stress where *F* is greater than zero. The threshold stress *K* is generally a scalar state variable and accounts for isotropic hardening (or softening). Note that the concept of a threshold function was introduced by Bingham [104] and later generalized by Hohenemser and Prager [105]. The threshold function is also referred to as the Bingham–Prager threshold function.

Taking the partial derivative of *Ω* with respect to *σ_ij_* as indicated in Equation (32) with the integral definition of *Ω* identified in Equation (39) leads to
(49)$$2\mu \frac{d\varepsilon_{ij}^{I}}{dt}=f\left(F\right)\sum_{ij}$$

Robinson [15] then specialized *f*(*F*) as a power law function
*f* (*F*) = *F^n^*
(50)
where *n* is an additional material constant. Thus,
(51)$$2\mu \frac{d\varepsilon_{ij}^{I}}{dt}=F^{n}\sum_{ij}$$

Similarly, taking the partial derivative of *Ω* with respect to the internal state variable *a_ij_* as indicated in Equation (49), the following evolutionary law is obtained for the change in state
(52)$$\frac{da_{ij}}{dt}=h\left(G\right)\frac{d\varepsilon_{ij}^{I}}{dt}-\left(\frac{R}{H}\right)h\left(G\right)g\left(G\right)a_{ij}$$

Robinson [15] then specialized *h*(*G*) as
(53)$$h\left(G\right)=\frac{H}{G^{\beta }}$$
and

*g* (*G*) = *G^m^*
(54)

Thus
(55)$$\frac{da_{ij}}{dt}=\left(\frac{H}{G^{\beta }}\right)\frac{d\varepsilon_{ij}^{I}}{dt}-\left(\frac{R}{H}\right)\left(\frac{H}{G^{\beta }}\right)G^{m}a_{ij}$$

Equation (55) is consistent with the physically based and well-accepted Bailey–Orowan theory [95,106]. The equation suggests that two competing mechanisms are present that control the evolution of the inelastic state. The first term represents a work-hardening mechanism that proceeds with inelastic strain. The second term represents a recovery process causing a softening that competes with the hardening term. When these two terms balance, the material attains a steady inelastic state.

Equations (51) and (55) comprise the multiaxial statement of the isotropic version of the Robinson viscoplastic constitutive model. The form of the flow potential function above is specialized to a *J*_2_ material. Several modifications, generalizations, and improvements of the original Robinson approach have been developed over the years, including the works by Duffy [107] and Saleeb et al. [108].

### 7.2. Chaboche’s Model

The Chaboche [22,23,24] model is a unified viscoplastic model also based on a potential function. This model includes isotropic and kinematic hardening state variables to capture the Bauschinger effect and cyclic hardening. Here, there are three internal state variables: a second-order tensorial back stress *α*_*ij*_ to account for kinematic hardening and the other two, yield strength *Y* and drag stress *K*, to account for isotropic hardening.
(56)$$f=\sqrt{\frac{3}{2}\left(S_{ij}-\alpha_{ij}\right)\left(S_{ij}-\alpha_{ij}\right)}-R-K$$

Note that *K* represents the yield stress, *R* is an isotropic hardening state variable, and *a_ij_* is a back stress (the kinematic state variable). As Chaboche [23] points out, there is a choice in the formulation of the viscoplastic potential *Ω* through this function *f*. Equation (56) is a *J*_2_ for *f*. Other tensorial invariants can be used to elicit different responses out of the model (see Duffy [107] and Saleeb et al. [108]). In general, the Chaboche [23] model is presented in a Norton [3] power law formulation as follows:(57)$$\Omega =\left(\frac{K}{n+1}\right){\left(\frac{f}{K}\right)}^{n+1}$$

Along with the deviatoric Cauchy stress introduced in the previous section, Chaboche [109] utilizes a deviatoric back stress
(58)$$a_{ij}=\alpha_{ij}-\frac{\alpha_{kk}\delta_{ij}}{3}$$

The inelastic strain rate is obtained from a viscoplastic potential function as follows:(59)$$\frac{d\varepsilon_{ij}^{I}}{dt}=\frac{\partial \Omega }{\partial \sigma_{ij}}\;\;=\frac{\partial \Omega }{\partial f}\frac{\partial f}{\partial \sigma_{ij}}\;\;=\lambda \frac{\partial f}{\partial \sigma_{ij}}$$
where the partial derivative of *f* with respect to *σ_ij_* is a gradient to the inelastic flow surface defined by *f*. The last form is recognized as a Prandtl–Reuss equation [110,111], where the multiplier *λ* is the magnitude of the inelastic strain rate vector. The direction of the gradient vector and the associated magnitude of the gradient vector are directly dependent on the choice of the formulation for *Ω* and *f*. Given the Norton power law formulation for *Ω* and a *J*_2_ yield function, the flow law takes the form
(60)$$\frac{d\varepsilon_{ij}^{I}}{dt}=\frac{K^{2}}{2\mu }{\left\{\frac{\sqrt{\frac{3}{2}\left(S_{ij}-\alpha_{ij}\right)\left(S_{ij}-\alpha_{ij}\right)}-R-K}{K}\right\}}^{n}\frac{\frac{3}{2}\left(S_{ij}-\alpha_{ij}\right)}{\sqrt{\frac{2}{3}\left[\frac{3}{2}\left(S_{ij}-\alpha_{ij}\right)\right]\left[\frac{3}{2}\left(S_{ij}-\alpha_{ij}\right)\right]}}$$

All three inelastic state variables evolve through competitive processes that include work hardening, deformation-induced dynamic recovery, and thermally induced state recovery. The evolution of the internal state can also include terms that vary linearly with the external variable rates. For isotropic hardening, the evolutionary law for the single-state variable (*K*) is
(61)$$\frac{dK}{dt}=b\overset{.}{p}\left(Q-R\right)$$
where
(62)$$\overset{.}{p}=\sqrt{\left(\frac{2}{3}\right)\frac{d\varepsilon_{ij}^{I}}{dt}\frac{d\varepsilon_{ij}^{I}}{dt}}$$
where *Q* and *b* are, respectively, the hardening saturation state and the rate of convergence to this state. These are parameters used to define the application of Chaboche’s model to strain range partitioning in fatigue.

For non-isothermal kinematic hardening,
(63)$$\frac{da_{ij}}{dt}=\left(\frac{2}{3}\right)C\left(\frac{d\varepsilon_{ij}^{I}}{dt}\right)-\gamma a_{ij}\overset{.}{p}-\frac{\gamma }{\tau \left(T\right)}{\left(\frac{\sqrt{a_{ij}a_{ij}}}{M}\right)}^{m-1}a_{ij}$$

For isothermal kinematic hardening,
(64)$$\frac{da_{ij}}{dt}=\left(\frac{2}{3}\right)C\left(\frac{d\varepsilon_{ij}^{I}}{dt}\right)-\gamma a_{ij}\overset{.}{p}$$

This is a formulation of an evolutionary law for a tensorial state variable that is similar to Robinson’s [15] given in Equation (52).

## 8. Viscoplastic Constitutive Models Not Based on Potential Functions

For models described in this subsection, the original formulations for the model were not presented as a derivation from a viscoplastic flow potential function. Subsequent presentations have included flow potentials for some of the models (see, for example, Kim and Oden [112]). Although functions are used to delineate elastic regions of the stress space from inelastic regions, the functions were not utilized to express a potential normality feature.

### 8.1. Bodner’s Model

Starting with Bodner’s early work [25], a major emphasis in the 1980s focused on combining the materials science flow rules for creep (specifically Norton’s [3]) with flow rules from plasticity. This effort forced different modeling approaches together, and internal state variable theory would join the concepts. Internal state variable theory gained influence as researchers embraced unified-creep-plasticity theories.

Bodner and Partom [26] proposed a flow rule with roots in classical plasticity and influenced by phenomenological observations, but which does not involve the specification of a yield surface. This theory is offered as a means of characterizing the isotropic hardening of certain materials. As in Chaboche’s model [109], where an invariant of the inelastic strain rate is utilized, Bodner et al. [32] adopt a Prandtl–Reuss [110,111] type of flow law:(65)$$\frac{d\varepsilon_{ij}^{I}}{dt}=\lambda S_{ij}$$

Squaring both sides of Equation (29) leads to
(66)$$\frac{d\varepsilon_{ij}^{I}}{dt}\frac{d\varepsilon_{ij}^{I}}{dt}=D_{2}^{I}\;\;=\lambda^{2}J_{2}$$
or the second invariant of inelastic strain rate is equal to the second invariant of deviatoric stress multiplied by the constant *λ*^2^, i.e.,
(67)$$D_{2}^{I}=\lambda^{2}J_{2}\;\;=F\left(J_{2},T,Z_{k}\right)$$

Here, *T* is temperature and *Z_k_* is a vector of inelastic state variables. One of the earliest forms for the flow law was based on a single state variable, Z, and was intended to capture isotropic hardening behavior. Specifically,
(68)$$D_{2}^{I}=D_{0}exp\left[-{\left(\frac{Z^{2}}{3J_{2}}\right)}^{n}\left(\frac{n+1}{n}\right)\right]$$
where *D*_0_ is a limiting inelastic strain rate in shear and *n* impacts strain rate sensitivity, as it does in all models rooted in Norton’s [3] creep law.

Bodner and Partom [27] postulated that the rate of change in a scalar-state variable was a function of inelastic work. For steady-state creep, there must be two competing state variables. Bodner’s model [113] is one of the earliest unified models that captures isotropic hardening through a nonrecoverable isotropic scalar state variable *Z^I^* and a directional second-order state variable for kinematic (directional) hardening *Z^D^*. These state variables combine linearly:

*Z* = *Z^I^* + *Z^D^*
(69)

The evolutionary law for the isotropic state variable is
(70)$$\frac{dZ^{I}}{dt}=m_{1}\left[Z_{1}-Z^{I}\left(t\right)\right]\frac{d{\overset{.}{W}}_{I}}{dt}-A_{1}Z_{1}{\left[\frac{Z^{I}\left(t\right)-Z_{2}}{Z_{1}}\right]}^{r_{1}}$$

Here, *m*_1_ is a hardening rate, and
(71)$$\frac{dW_{I}}{dt}=\sigma_{ij}\left(\frac{d\varepsilon_{ij}^{I}}{dt}\right)$$
is the inelastic work rate and is a measure of work hardening. The negative part of Equation (69), representing the first state variable, can be interpreted as dynamic recovery. It is a thermal or “static” recovery of hardening where *Z*_2_ is the stable (minimum) value of *ZI* at a given temperature and *A*_1_ and *r*_1_ are temperature-dependent material constants.

Kinematic (directional) hardening is represented by a second-order symmetric tensor *β_ij_* and a second order tensor of direction cosines from the current state of stress, i.e., *Z^D^* takes the form

*Z^D^* (*t*) = *β_ij_* (*t*) *d_ij_* (*t*)
(72)

### 8.2. Walker’s Model

The Walker model [50] was originally developed in an integral form by modifying the constitutive relation for a three-parameter viscoelastic solid. Walker [50] points out that later a differential form of the model was proposed, and this differential format is presented here. Two state variables *ω_ij_* and *K* were introduced into the viscoelastic model to account for kinematic (directional) and isotropic hardening. The back stress *ω_ij_* accounts for kinematic hardening and Bauschinger effects. The isotropic state variable *K* affects cyclic hardening or softening in a material undergoing repeated loads.

The Walker model [50] is composed of four rate equations and an exponential function. The formulation for the inelastic strain rate depends on the applied deviatoric stress tensor, *S_ij_*, and the two state variables *ω_ij_* and *K* in the following manner
(73)$$\frac{d\varepsilon_{ij}^{I}}{dt}={\left\{\frac{\sqrt{\frac{2}{3}\left(\frac{3S_{ij}}{2}-\omega_{ij}\right)\left(\frac{3S_{ij}}{2}-\omega_{ij}\right)}}{K}\right\}}^{n}\frac{\frac{3S_{ij}}{2}-\omega_{ij}}{\sqrt{\frac{2}{3}\left(\frac{3S_{ij}}{2}-\omega_{ij}\right)\left(\frac{3S_{ij}}{2}-\omega_{ij}\right)}}$$

The evolutionary law for the back stress state variable is stipulated as
(74)$$\frac{d\omega_{ij}}{dt}=\left[\left(n_{1}+n_{2}\right)+\frac{\partial n_{1}}{\partial \theta }\frac{\partial \theta }{\partial t}\right]\frac{d\varepsilon_{ij}^{I}}{dt}-\left(\omega_{ij}-{\overset{\sim }{\omega }}_{ij}-n_{1}\varepsilon_{ij}^{I}\right)\left[\frac{dG}{dt}-\frac{1}{n_{2}}\frac{\partial n_{2}}{\partial \theta }\frac{\partial \theta }{\partial t}\right]$$
where *n*_1_, *n*_2_, and $\overset{}{\overset{\sim }{\omega_{ij}}}$ are material constants. Here, *Θ* represents temperature, and *G* is a recovery function. Although not given as a rate equation, the current value of the state variable *K* depends on the norm of the inelastic strain rate
(75)$$\frac{dR}{dt}=\sqrt{\frac{2}{3}\frac{d\varepsilon_{ij}^{I}}{dt}\frac{d\varepsilon_{ij}^{I}}{dt}}$$
as follows
*K* = *K*_1_ − *K*_2_ exp(−*n*_3_*R*)
(76)

In Equation (74), the time rate of the change in the recovery function *G* is
(77)$$\frac{dG}{dt}=\left\{n_{4}+n_{5}\left[exp\left(-n_{6}R\right)\right]\frac{dR}{dt}\right\}+n_{7}{\left(\frac{2\omega_{ij}\omega_{ij}}{3}\right)}^{\frac{m-1}{2}}$$

The model parameters *n*_3_, *n*_4_, *n*_5_, *n*_6_, *n*_7_, *K*_1_, *K*_2_, and *m* are material constants. All material constants in this model are considered temperature-dependent. See Cassenti [114] for details regarding temperature effects impacting the constants in the Walker model [50], the Miller model [36], and the Krieg, Swearengen, and Rhode model [115].

### 8.3. Miller’s Model

The Miller [36,37,38,39,40,41,42,43,44,45,46,47,48] viscoplastic model captures a wide range of physical phenomena, e.g., cyclic hardening, softening, and saturation behavior. Walker [50], Allen and Harris [53], and Cassenti [114] point out that in the original Miller model, the inelastic strain rate was formulated as
(78)$$\frac{d\varepsilon_{ij}^{I}}{dt}=B\theta^{\prime }{\left\{sinh{\left[\frac{\sqrt{3J_{2}^{\prime }}}{K}\right]}^{\frac{3}{2}}\right\}}^{n}\frac{\left(\frac{3S_{ij}}{2}-\omega_{ij}\right)}{\sqrt{3J_{2}^{\prime }}}$$
and the rate of change for the kinematic state variable is
(79)$$\frac{d\omega_{ij}}{dt}=H_{1}\frac{d\varepsilon_{ij}^{I}}{dt}-H_{1}B\theta^{1}\left\{{\left[A_{1}\sqrt{\frac{2\omega_{ij}\omega_{ij}}{3}}\right]}^{n}\right\}\frac{\omega_{ij}}{\sqrt{\frac{2\omega_{ij}\omega_{ij}}{3}}}$$

In these two expressions, *B*, *A*_1_, and *H*_1_ are material constants. The rate of change in the scalar state variable is
(80)$$\frac{dK}{dt}=H_{2}\frac{dR}{dt}\left\{{\left[C_{2}+\sqrt{\frac{2\omega_{ij}\omega_{ij}}{3}}-\left(\frac{A_{2}}{A_{1}}\right)K\right]}^{n}\right\}-H_{2}C_{2}B\theta^{\prime }{\left[sinh\left(A_{2}K^{3}\right)\right]}^{n}$$

In this last expression, *A*_2_, *C*_2_, and *H*_2_, are material constants, and
(81)$$\begin{array}{ll} \theta^{\prime }=exp\left(-\frac{Q^{*}}{kT}\right) & for\;T\geq 0.6T_{m}\\ \theta^{\prime }=exp\left(-\frac{Q^{*}}{kT}\right)ln\left[1+\frac{0.6T_{m}}{T}\right] & for\;T<0.6T_{m} \end{array}$$
just as in the Walker [50] model,
(82)$$\frac{dR}{dt}=\sqrt{\frac{2}{3}\frac{d\varepsilon_{ij}^{I}}{dt}\frac{d\varepsilon_{ij}^{I}}{dt}}$$

The hyperbolic sine function was adopted in this model to capture the creep responses more accurately in a material. Hence, the hyperbolic sine formulation is maintained in the derivation of the thermal-recovery variables of back-stress and drag-stress evolution laws. This particular unified model possesses the ability to predict the Bauschinger effect, history dependence, temperature dependence, anelasticity, and multiaxial response phenomena.

### 8.4. Krieg, Swearengen, and Rhode’s Model

The model developed by Krieg et al. [115] is one of the first to incorporate both a drag-stress and a back-stress term in a viscoplastic model. The inelastic strain rate is identical to the expression in Walker’s model [50], i.e.,
(83)$$\frac{d\varepsilon_{ij}^{I}}{dt}={\left\{\frac{\sqrt{\frac{2}{3}\left(\frac{3S_{ij}}{2}-\omega_{ij}\right)\left(\frac{3S_{ij}}{2}-\omega_{ij}\right)}}{K}\right\}}^{n}\frac{\frac{3S_{ij}}{2}-\omega_{ij}}{\sqrt{\frac{2}{3}\left(\frac{3S_{ij}}{2}-\omega_{ij}\right)\left(\frac{3S_{ij}}{2}-\omega_{ij}\right)}}$$

However, the evolutionary law for the kinematic state variable is different:(84)$$\frac{d\omega_{ij}}{dt}=A_{1}\frac{d\varepsilon_{ij}^{I}}{dt}-A_{2}\omega_{ij}\sqrt{\frac{2\omega_{pq}\omega_{pq}}{3}}\left[exp\left(\frac{2A_{3}\omega_{pq}\omega_{pq}}{3}-1\right)\right]$$

The current value of the isotropic state variable *K* is given by the expression
(85)$$K=A_{4}\frac{dR}{dt}-A_{5}{\left(K-K_{0}\right)}^{n}$$

The value for *K* depends on the norm of the inelastic strain rate as follows:(86)$$\frac{dR}{dt}=\sqrt{\frac{2}{3}\frac{d\varepsilon_{ij}^{I}}{dt}\frac{d\varepsilon_{ij}^{I}}{dt}}$$

No explicit provision exists to model cyclic hardening in either of the state variables. The constants *A*_1_ through *A*_5_ appearing in the growth laws for the inelastic strain rate and the evolutionary laws for the internal state variables do not explicitly depend on the cumulative inelastic deformation. The linear hardening terms and the recovery term in the state variable evolutionary equation yield stress–strain curves and hysteresis loops that exhibit the same tri-linear character as Miller’s theory.

### 8.5. Hart’s Model

The Hart [116,117,118] viscoplastic model is one of the early unified creep-plasticity models based on internal state variables. Hart [118] took a microstructural viewpoint in developing his continuum model. He was intent on modeling creep behavior based on experimental evidence of the existence of barriers to dislocations and suggested the existence of a barrier where dislocations “piled up” and a resistance or friction opposing dislocation movement within barriers. Here, the relation between the applied stress, the internal stress, and the glide friction stress was derived as the internal stress was shown to be linearly proportional to a stored anelastic strain. Hart’s model [118] is unique because the isotropic state variable, typically represented as *K* in other viscoplastic models, is constant. Since this parameter is a constant in the Hart [118] model, it cannot be considered an internal state variable.

The inelastic strain rate is similar to the expression in Walker’s model [50], i.e.,
(87)$$\frac{d\varepsilon_{ij}^{I}}{dt}=a^{*}{\left\{\frac{\sqrt{\frac{2}{3}\left(\frac{3S_{ij}}{2}-\omega_{ij}\right)\left(\frac{3S_{ij}}{2}-\omega_{ij}\right)}}{K}\right\}}^{n}\frac{\frac{3S_{ij}}{2}-\omega_{ij}}{\sqrt{\frac{2}{3}\left(\frac{3S_{ij}}{2}-\omega_{ij}\right)\left(\frac{3S_{ij}}{2}-\omega_{ij}\right)}}$$
where *S_ij_* is the applied deviatoric stress tensor and *ω_ij_* is a kinematic internal state variable. The parameter *a** is a material constant. The evolutionary law for the kinematic internal state variable is
(88)$$\frac{d\omega_{ij}}{dt}=K\left\{\frac{d\varepsilon_{ij}^{I}}{dt}-f\left\{\frac{{\left(\sigma^{*}\right)}^{m}\omega_{ij}}{\sqrt{\frac{2\omega_{pq}\omega_{pq}}{3}}{\left[ln\left(\frac{\sigma^{*}}{\sqrt{\frac{2\omega_{pq}\omega_{pq}}{3}}}\right)\right]}^{\frac{1}{\beta }}}\right\}\right\}$$

The variable *σ** could be considered a secondary scalar state variable since it serves only to modify the equilibrium stress state variable *ω_ij_*. The rate of change of this scalar variable is given by the following expression:(89)$$\frac{d\sigma^{*}}{dt}=\frac{Cf{\left(\sigma^{*}\right)}^{m+1}⟌\left(\sigma^{*},\omega_{ij}\right)}{ln{\left(\frac{\sigma^{*}\omega_{pq}\omega_{pq}}{\sqrt{\frac{2}{3}}}\right)}^{\frac{1}{\beta }}}$$

Note that
(90)$$⟌\left(\sigma^{*},\omega_{ij}\right)=\left(\frac{\gamma }{\sigma *}\right)$$
although Delph [119] offered other forms for Equation (94). In the above expressions, *λ*, *μ*, *a**, *K*, *n*, *f*, *σ*_0_***, *β*, *C*, γ, and δ are material constants that depend on temperature. In addition, *σ*_0_*** is the initial value of *σ*.*

## 9. Integral-Based Viscoplastic Models

The previous viscoplastic models are examples of representing nonlinear constitutive models using a differential equations format. Whereas differential representation is presented in the stress space, the integral representation of material behavior is conducted in the strain/deformation space. As Walker [50] points out, several previous viscoplastic models have both differential and integral formulations. The endochronic models focus on the inelastic response of materials through the use of memory integrals known as memory kernels. Early on, Bouc [120,121] investigated the mathematics of hysteresis, which led to the study of a class of applicable functional operators. In structural engineering, the Bouc–Wen [121,122] model of hysteresis is typically employed to describe nonlinear/inelastic hysteretic systems. In the context of engineering mechanics and structural dynamics, the Bouc–Wen [121,122] model is a phenomenological model that captures both the linear–elastic and elasto-plastic restoring forces in systems that exhibit hysteretic phenomena.

The Bouc–Wen [121,122] model is widely employed for modeling the cyclic behavior of structures in seismic engineering. Specifically, jointed connections dissipate energy through friction over localized regions near the connection and are known to exhibit history-dependent or hysteretic behavior. The Bouc–Wen [121,122] model and subsequent variations are rate-independent hysteresis models. The hysteresis model proposed by Valanis [87] is a rate-dependent hysteresis model. Both the Bouc–Wen [121,122] and Valanis [87] models are grounded in the mechanical modeling concepts found in viscoelasticity. Viscous damping can also be formulated by describing the damping force as a function of the cumulative history of the system, i.e., memory effects. Endochronic models proposed by Valanis [87] and the Bouc–Wen [121,122] are two important examples of hereditary models in endochronic theory that are used in modeling nonlinear material behavior. The Valanis [87] model and its extension to viscoplasticity behavior is focused on here.

### Valanis Model

The main principle of a hysteretic model is known as intrinsic time, which relates the deformation history to a deformation memory component. Initially, the endochronic theory introduced by Valanis [87] described a nondecreasing function dependent on the strain tensor *ε_ij_* or the stress tensor *σ_ij_*. The original Valanis [87] endochronic model is built on a single integrated framework and is independent of a yield surface. Resembling the viscoelastic theory, the endochronic theory replaces real time with an auxiliary time variable, referred to as intrinsic time. The constitutive theory developed on the endochronic theory characterizes the hysteresis and strain-hardening behavior of some metals independent of yield conditions, flow rule, or any hardening rules. The intrinsic time-dependent stress evolution rule for the endochronic theory is derived by a convolution integral between the strain tensor *ε_ij_* and the scalar function called a memory kernel. According to endochronic theory, the time-dependent constitutive relationship is defined as
(91)$$\sigma_{ij}\left(t\right)=\delta_{ij}\left(\lambda +\frac{2\mu }{3}\right)\varepsilon_{kk}\left(t\right)+\int_{0}^{t}G\left[z\left(t\right)-z\left(\xi \right)\right]\left\{\frac{\partial \varepsilon_{ij}}{\partial \xi }-\frac{\delta_{ij}}{3}\frac{\partial \varepsilon_{kk}}{\partial \xi }\right\}d$$

When the memory kernel *G* is an exponential function, an incremental form of endochronic flow rules exists. The exponential formulation is commonly used in standard deformation analyses and applications. In general,
(92)$$G\left[z\left(t\right)-z\left(\xi \right)\right]=G_{1}exp\left\{-a_{1}\left[z\left(t\right)-z\left(\xi \right)\right]\right\}+G_{2}exp\left\{-a_{2}\left[z\left(t\right)-z\left(\xi \right)\right]\right\}$$

The function *z*(*t*) is defined by the following expression
(93)$$z\left(t\right)=\frac{1}{\beta }ln\left(1-\beta R\left(t\right)\right)$$
where
(94)$$R\left(t\right)=\int_{0}^{t}\left[\frac{\partial \theta }{\partial \xi }f\left(\frac{\partial \theta }{\partial \xi }\right)\right]d\xi$$
and
(95)$$\frac{\partial \theta }{\partial \xi }=\sqrt{\frac{2}{3}\frac{\partial c_{ij}}{\partial \xi }\frac{\partial c_{ij}}{\partial \xi }}$$

Finally,
(96)$$\frac{\partial c_{ij}}{\partial \xi }=\partial_{ij}\lambda \frac{\partial \varepsilon_{kk}}{\partial \xi }+2\mu \frac{\partial \varepsilon_{ij}}{\partial \xi }-k\frac{\partial \sigma_{ij}}{\partial \xi }\;0<k<1$$
And in the expression above, *λ*, *μ*, *G*_1_, *G*_2_, *a*_1_, *a*_2_, *β*, and *k* are material constants.

Several authors, including Bazant and Bhat [90], developed methods of extending endochronic theory to concrete, clay, and sand using the Valanis [87] model as a point of departure. This was carried out by introducing intrinsic time to the hydrostatic component of the stress state and incorporating inelastic dilatancy by way of the shear strain. The Valanis [86,87,88,89] approach to modeling is useful in predicting the nonlinear material behavior of metal-forming crucibles obtained from a powder-metal-forming process. Locating a threshold surface for a high-temperature powder metal can be difficult if the powder metal material is not relatively close to being fully dense. The endochronic theories [123] can be useful from the standpoint that a yield surface is not required to perform an analysis. A subsequent version of the endochronic theory was developed by Valanis [124], where the intrinsic time was redefined as the path length in the inelastic strain space. The new endochronic model could predict a temporal stress response for several deformation processes. Wu and Ho [125] introduced another functional form for the dependency upon the intrinsic time scale to the equivalent deviatoric plastic strain rate and applied endochronic theory to investigate the transient creep of material.

## 10. Design Scale

In the preceding discussions, modeling inelastic deformation behavior has been presented primarily at a continuum level with references to the physics at the microstructural/mesoscale. Historically, components have been designed at the continuum level, and that design viewpoint still dominates today. The continuum-level design approach is taught to all undergraduate design engineers. However, the continuum-level models must have a basis in physics and a connection to the mesoscale models that have been presented. If that connection to physics is not there, then the continuum-level model is a curve fit. Moreover, if the goal is to improve material performance, then one designs the material at the microstructural level.

The different microstructures within a material (grain morphology, size distributions, anisotropy, and crystallographic orientation; the presence of flaws and porosity; the physical and chemical properties of the intergranular interfaces, etc.) directly influence the performance properties of a material. A link between microstructure and continuum can provide valuable insight into the design of components fabricated from high-performance materials. It should be noted that the design perspective at which a material/component/system is modeled strongly depends on the experimental information available. Internal state variable (ISV) concepts have been formalized as a continuum-level parameter that accounts for dislocation mechanics. This provides a link to the prediction of the movement of dislocations through a microstructure. System-level parameters such as flexibility and stiffness coefficients are based on continuum-level parameters and geometry.

Relative to time-dependent behavior, constitutive models are available at both levels with convenient bridges identified by researchers over the years. In the next section, a discussion is provided on those links to different design scales.

## 11. The Bridge from Microstructure to Continuum: Time-Dependent Behavior

ISVs account for local unified creep and plasticity mechanisms driving time-dependent deformation behavior at the microstructural level. Coleman and Gurtin [126] connected the behavior of a material under applied load at the microstructural level with the incorporation of the internal state variables in the material model. Lubliner [127,128,129] also showed that a material with a property of fading memory can be described with internal state variables. The fading memory property refers to the hypothesis that the values of stress are dependent on recent deformations when compared to the deformations incurred in the distant past.

To capture the damage in materials, Horstemeyer et al. [130] postulated an ISV theory combining the hardening equations and porosity evolution equations. They pointed out how a finite element model that explicitly incorporates ISVs can be used to capture the structure–property relations in the realm of large deformations. A finite element simulation accommodates different grain sizes, particle sizes, pore sizes, and volume fractions within each element so that a material with a heterogeneous microstructure can be represented within the whole mesh. Based on the plasticity/damage approach proposed by Bamman et al. [131], Horstemeyer et al. [130] made improvements to the theory by adding ISV equations that introduce grain particle size, particle volume fraction, grain size, pore volume fraction, and nearest neighbor distances of the particles and pores. This allowed the use of heterogeneous distributions of microstructures throughout a finite element mesh. This multiscale modeling approach demonstrated that the standard assumption of homogeneous distributions of microstructural features such as porosity (i.e., the classical approach) can lead to erroneous results and conclusions. This is the benefit of using microstructural property-based ISV theories in conjunction with finite element analysis.

The deformation process in a material generally involves nonuniform fields present at the microstructural level. As noted earlier, Rice [132] examined the relationship between inelastic strain and slip displacements. He concluded that the slip within individual grains may be considered a deformation in the usual continuum sense and that each point of the inelastic strain can be broken down into a finite set of simple shearing strains of the respective slip planes and directions.

Thus, at the microstructural level, the material can be considered an ensemble of distinct entities, e.g., individual grains in polycrystalline metals. Many of the early continuum-level viscoplastic models adopted the Rice [132] interpretation to make a connection between the continuum level and the microstructure.

One must be cognizant of the material evolution caused by dislocation motion taking place in the material microstructure, and this can be achieved by studying well-accepted material science models. Blum and Eisenlohr [133] point out that insight obtained from material science models relative to how dislocation structures and microstructural morphology impact deformation behavior should be reflected in continuum-level models. However, material science models tend to not paint a complete picture in the sense that primary creep (the transient response) is captured in some material science models and ignored in other models that focus on steady-state response. Material science models are typically not posed relative to multiaxial stress states. Several schools of thought exist on the topic of representing material behavior in a fully three-dimensional setting with microstructural-level information explicitly incorporated. Consider an approach that employs finite element analysis to develop a representative volume element (RVE) that captures microstructural level features. Hill [134] first suggested how to construct models that can be constructed based on these RVEs that analyze components at the continuum level in a fully three-dimensional setting. The RVE approach has its merits in that the physics of dislocation mechanics is captured. Modeling effort using the RVE approach requires an interpretation of boundary conditions, as well as knowledge of initial dislocation densities.

The ability to observe, characterize, simulate, and then design components at the mesoscale level requires advances in techniques. For example, to observe how the microstructure of the test specimen evolves, High Energy X-ray Diffraction Microscopy (HEDM) is the frontrunner because of its ability to look at the mesoscale structure. This ability to see with microscale resolution helps in characterizing the polycrystal orientation and strain states under thermal and mechanicals loads. There is a need to characterize the initial dislocation field in situ to model from the microstructural level up to the continuum level. The challenge is in defining the right boundary conditions; in the case of a single crystal with free surfaces, free boundaries (dislocations are free to vanish when they reach the surface) are defined or closed boundaries (the dislocations can penetrate the surface) are defined to capture the behavior of a polycrystal. At the microstructural level, the dislocations used to define the initial dislocation density are randomly distributed with respect to their length and number. These definitions, integrated over the simulated region, can be used to derive the global quantities, e.g., the stress–strain curves, dislocation densities, and local qualities calculated at every step of the simulation. In some regions, these stress states must be allowed to vary, where the trajectories of the dislocation path and the glide plane are tracked.

There is a need to bridge the two scales and master the art of assembling structural and functional microstructural models into larger continuum models that allow for the design of complex structural components. Polycrystalline materials are among the most important due to their ability to distribute load via plastic deformation. The two main mechanisms through which plastic strains are expressed are the flow of dislocations and twinning, both of which are characterized as in-line defects in crystals. These mechanisms are understood individually, but their interactions that affect the behavior at the mesoscale level remain a challenge. For example, elastic moduli are calculated using first principles but the computation for stress–strain curves, as required by engineers, is a challenge because of the complicated dislocation mechanics at the atomistic level.

The phenomenological models discussed in earlier sections are continuum-level deformation models. The phenomenological constitutive models based on dislocation theories incorporate a temporal relationship between continuum level stress and strain. Currently derived phenomenological models do not explicitly account for the interaction between material microstructure and the continuum. However, well-thought-out phenomenological models should in some fashion mirror the evolution of the material at the microstructural level mathematically. Phenomenological models may not explicitly capture in detail the material behavior within the microstructure. However, the question should be asked as to what level should a component be designed at. Should one start the design of components at the atomic/molecular level, or should one appeal to nanomechanics/micromechanics, or the mesomechanics continuum level? Historically, engineering design has been cognizant of material behavior at the microstructural level, but component designs are executed at a continuum level. To begin a discussion on bridging these schools of thought, a brief overview is provided regarding the details supporting representative volume elements (RVE).

There are always gaps between the modeling efforts at one level relative to the next level. A prevalent philosophy has been that the design engineer designs components using phenomenological models and continuum mechanics, while material scientists design the material with microstructural models. The modeling at both levels should have a symbiotic relationship. Bodner and Partom [27] attempted to bridge the gap between the material microstructure and the continuum by including certain physical concepts supporting dislocation mechanics. Others followed by similarly pointing to physical mechanisms that their continuum models capture. Here, a transitional approach is discussed where the physics of the microstructure can be captured by a representative volume element. Instead of pointing to microstructural mechanisms and attempting to mirror the mechanics, the intent is to explicitly incorporate the mechanisms via a discretized finite element mesh. This approach evolved with the intent to develop a collaboration between material science and engineering mechanics. Mesomechanics allows the adoption of a heterogeneous medium with a distinct microstructure and micromechanics that evolve the microstructure as opposed to a continuum with averaged properties. The limitations associated with mesomechanics include the need for a precise description of the microstructure and the inability to capture the complexities in the evolution of complex material structures.

Non-linear deformations and time-dependent material behavior based on the concept of representative volume element were first adopted by Hill [134]. Since finite element analysis is based on the continuum, mechanics and small deformations are typically used under the assumption of linear deformations. To circumvent the limitations of heterogeneity and linear processes, a few methods have been proposed in the form of asymptomatic homogenization theory by Terada and Kikuchi [135] and Ghosh et al. [136,137] and volume averaging by Smit et al. [138] and Feyel and Chaboche [139]. In contrast to assigning homogenized constitutive behavior at integration points, the average behavior associated with the microstructure is assigned at each integration point of the macromesh of the representative volume element (RVE).

To achieve precise detail at the microstructural level and to better understand the link between microstructure and continuum, a polycrystal plasticity finite element model was proposed by Holm et al. [140]. The authors addressed the limitations of classical constitutive models by allowing an atoms-up approach that also produces a continuum down the path. Their model includes geometry for pores as well as the grain structure. The authors use a paradigm where a combined computational and experimental assessment of the microstructure leads to the homogenization of the material. The homogenized properties are then utilized in continuum level engineering design. Digital image correlation coupled with electron-backscatter diffraction microscopy provide strain maps that can validate microstructure. Characteristic features and their distributions and trends are revealed that validate simulation results, e.g., plastic strain bands that appear oriented at 45° in components subjected to uniaxial loads.

A discrete element method proposed by Horner et al. [141], along with Iwashita and Oda [142] in collaboration with Tordesillas [143], suggests the adoption of the smallest possible representative volume element (RVE) to ensure high resolution at the microstructural level to capture the dynamics and kinematics of the particles to link to the evolution of the microstructural mechanics.

The micromorphic continuum theory is outlined in Mindlin [144], where the material is observed as a collection of deformable “points” used to model material parameters and generate models at the nanoscale. This was accomplished by using numerous bridging scale methods in conjunction with continuum, quantum, and molecular mechanics as demonstrated in Liu et al. [144,145]. To further the application of these methods, a new theory called the multiresolution continuum theory was proposed by Liu et al. [144,145] for heterogeneous materials. Gao et al. [146] expanded the micromorphic theory by incorporating geometric relevant material defects.

Finally, by incorporating damage mechanics at the microstructural level in the classical continuum models also helps bridge the gap and capture the material behavior at several sub-levels, as was proposed in the work of Holm [147], who continued the development of future multiscale modeling theories by adopting various approaches. The upscaling method and resolved-scale method are reviewed in Fish et al. [148]. Holm et al. [148] state that understanding various processes of microstructural scale damage where the interaction between the grain boundary with dislocations and twins is important and critical in modeling damage mechanics. Continuum models fail to predict fracture. The goal is that the microscale damage models can bridge this gap. Methods have been developed to bridge the gap across length and time scales, as discussed in Fish et al. [149].

## 12. Concluding Remarks

For over a century, the materials community has been proposing methods to characterize the nonlinear stress–strain behavior exhibited by materials. In this review article, a collection of works is presented that depicts the development of inelastic constitutive models that have applications at the continuum level, a homogenized polycrystalline material. The discussion begins with classical plasticity theories and that discussion carries through to time-dependent viscoplastic models.

Focus has been given to material science models that physically explain nonlinear behavior at the microstructural level. An understanding of material microstructure is always necessary in developing accurate multiaxial continuum-level constitutive models that characterize the responses of engineering components modeled with continuum-level perspectives.

For further reading, attention is drawn to the work of Vladmir Buljak and Gianluca Ranzi [149], which captures the distinction between various inelastic constitutive models from a different perspective.

A bridge between microstructural concepts and continuum concepts is convenient. This bridge is discussed in the context of time-dependent behavior constitutive models.

Many forms of inelastic constitutive models are presented that include differential formulations as well as integral forms. Models are available that aid in the design of components that function in high-temperature environments. This would be accomplished by incorporating three-dimensional continuum-level inelastic models in finite element analysis software. A discussion regarding that effort is available elsewhere.

## Figures and Tables

**Figure 1 materials-16-03564-f001:**
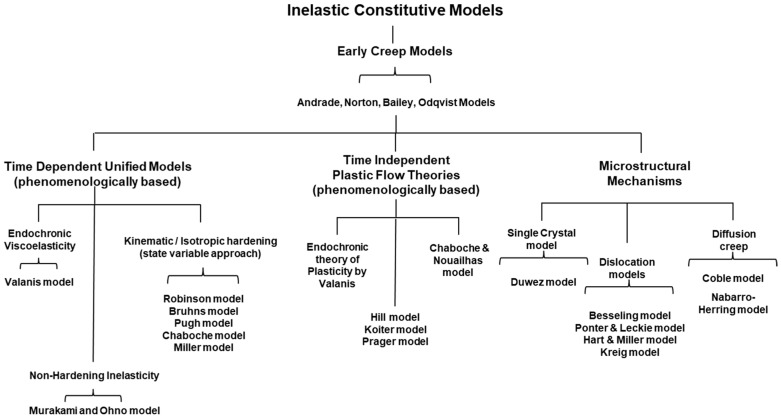
Classification of Inelastic Constitutive Models.

**Figure 2 materials-16-03564-f002:**
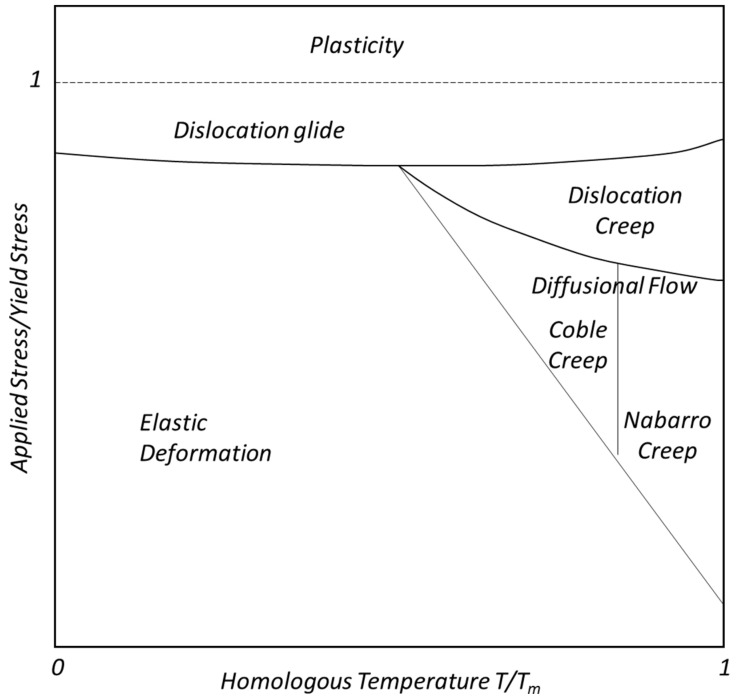
Generic deformation mechanism map.

## Data Availability

Not applicable.

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
