# Peer review of "Review: Inelastic Constitutive Modeling: Polycrystalline Materialsâ€"

_materials, 2023, doi:10.3390/ma16093564_

Round 1
Reviewer 1 Report
The manuscript entitled "Review: Inelastic Constitutive Modelling – Polycrystalline Materials" reviews the details of the development of inelastic constitutive modeling as it relates to polycrystalline materials. The review is well-written and easy to understand. My specific comments on the article are given as follows
1. Some works done in the field in recent years such as in 2022, 2021, and if any 2023 should also be cited in the article.
2. There are some typographical errors which are needed top be corrected
Author Response
Response is given in the attached word document.

Reviewer 2 Report
This paper provides a comprehensive literature review of the development and application of inelastic constitutive models for polycrystalline materials. The authors distinguish between different types of models used to account for nonlinear behavior at microstructural levels, including time-independent classic plasticity models and time-dependent unified viscoplasticity models.
One key contribution is that particular emphasis was placed on understanding the theoretical framework underlying these approaches with respect to creep and classical plasticity behaviors resulting from applied boundary conditions rather than separable rates representing distinct physical mechanisms. By providing clarity around these concepts, researchers can develop more accurate multiaxial continuum-level constitutive models based on material microstructures.
Overall, while there are limitations associated with any specific approach or model discussed in this literature review due to differences among various materials or structures being modeled; In general, the submitted manuscript could be suitable for publication in "Materials".
Author Response

(The authors gave the same response as above.)

Reviewer 3 Report
In this investigation, the development of inelastic constitutive models for polycrystalline materials during forming is detailed reviewed. The results are interesting and theoretical significance. Several words and grammar can be improved.
(1) Authors focused on summarizing the modeling principles of various models. However, the prediction accuracies of different models were not analyzed. Why?
(2) The format of the reviewed paper needs to be improved. For example, the section number is missing.
(3) The words and grammar in present study should be further improved.
Author Response

(The authors gave the same response as above.)

Reviewer 4 Report
1. The reviewer come across some grammar errors which need be deleted.
2. Although many works have been covered, some works in recent 5 years should be added.
3. Some comparative works are lacked in this paper.
4. The specific background of inelastic constitutive modeling are expected to be added.
5. It is better to include some simulation results.
Author Response

(The authors gave the same response as above.)

Reviewer 5 Report
The Authors present a wide review of the development of inelastic constitutive modeling as it relates to the polycrystalline materials.
Careful editing and proofreading of the complete manuscript are advisable to correct some mistyping. For example: " … slip with irradiationenhanced climb," (page 9, line 41), "... a full threedimensional setting " (page 37, line 5), and others
Author Response

(The authors gave the same response as above.)
